# The imprint of star formation on stellar pulsations

Thomas Steindl ◉[1] ✉, Konstanze Zwintz[1] & Eduard Vorobyov ◉[2,3]

In the earliest phases of their evolution, stars gain mass through the acquisition of matter from their birth clouds. The widely accepted classical concept of early stellar evolution neglects the details of this accretion phase and assumes the formation of stars with large initial radii that contract gravitationally. In this picture, the common idea is that once the stars begin their fusion processes, they have forgotten their past. By analysing stellar oscillations in recently born stars, we show that the accretion history leaves a potentially detectable imprint on the stars' interior structures. Currently available data from space would allow discriminating between these more realistic accretion scenarios and the classical early stellar evolution models. This opens a window to investigate the interior structures of young pulsating stars that will also be of relevance for related fields, such as stellar oscillations in general and exoplanet studies.

In recent years, asteroseismology—the analysis of stellar oscillations—has revolutionised our understanding of galactic and stellar evolution and has become one of the most powerful tools to unravel the interior structures of different types of stars from the core-hydrogen burning stage to the final phases of evolution (e.g. white dwarfs). Since oscillation modes propagate throughout the stellar interior, their frequencies contain information about the inner structure of the stars. Thanks to phenomenal data obtained from space telescopes such as MOST[1], CoRoT[2], BRITE-Constellation[3], Kepler[4] and TESS[5], the investigation of the oscillation properties of stars in a mass range from about 0.5 to ~40 $M_\odot$ with effective temperatures between about 3000 and 100,000 K allowed to test and improve our current theories of stellar structure and evolution[6]. Multiple studies have shown that the uncertainty of theoretical frequencies is nowadays higher than the uncertainty on observationally inferred pulsation frequencies, given that the time-base of observations is long enough.

Asteroseismology of stars before the onset of core-hydrogen burning, i.e. pre-main sequence stars, is a comparably young research field that has advanced due to dedicated observations[7] and adaptations of theoretical model calculations[8]. However, our general theory of early stellar evolution lacks essential physical ingredients that are either not well understood or not accurately included in our

theoretical models. As the earliest phases in the lives of stars determine their future fate and the creation and evolution of planetary systems, an improved understanding of these stages is essential.

The often adopted classical picture of early stellar evolution assumes a fully convective star with a large initial radius. In this case, a two solar mass star has an initial radius of 55 solar radii when creating a pre-main sequence model according to our standard input physics (see methods, subsection stellar evolution models) with the standard routine of MESA[9], which subsequently contracts along the Hayashi track[10]. The contraction releases energy from the gravitational potential, heating the stellar interior and fueling the convection. With higher temperatures, the onset of thermonuclear reactions slows the contractions, the stellar interior becomes radiative and the subsequent evolution of the young star in the Hertzsprung-Russell diagram follows the Henyey track[11]. Further contraction then leads the stellar core to burn the initially present $^{12}C$ into $^{14}N$ via the first parts of the CNO cycle ($^{12}C(p, \gamma)^{13}N(\beta+, \nu)^{13}C(p, \gamma)^{14}N$), which is a chain of nuclear reactions catalysed by carbon (C), nitrogen (N) and oxygen (O). The star further continues to contract towards the Zero Age Main Sequence (ZAMS) only after the initial $^{12}C$ has been lowered to equilibrium abundances, leading to additional heating until the full CNO cycle can operate in equilibrium. While energy production of the CNO cycle dominates for

[1]Institute for Astro- and Particle Physics, University of Innsbruck, Technikerstraße 25, A-6020 Innsbruck, Austria. [2]Department of Astrophysics, University of Vienna, A-1180 Vienna, Austria. [3]Institute of Astronomy, Russian Academy of Science, 48 Pyatnitskaya St., Moscow 119017, Russia. ✉e-mail: thomas.steindl@uibk.ac.at

stars discussed in this work, stars with masses below ∼1.3 solar masses obtain most of their energy from the proton-proton chain. This marks the onset of core-hydrogen burning, the start of the stars' main sequence lifetime and, with it, the end of the pre-main sequence evolution. The classical concept of stellar evolution assumes further that at the time the stars arrive on the main sequence, their previous stages do not leave a traceable imprint: the stars forget the details of their early evolutionary history.

This simplified picture of early stellar evolution has been overhauled by multiple authors in the last decades[12–20] but has so far not reached the field of asteroseismology. This includes the wrong but deep-rooted view that stars suddenly become optically visible once passing the stellar birthline[16]. In contrast to the classical pre-main sequence evolution models that simplified initial conditions, state-of-the-art calculations of these earliest phases in stellar evolution start from protostellar seeds with a few Jupiter masses and about one solar radius. The ZAMS mass is then reached through accretion throughout the first few million years. The evolutionary tracks of accreting protostars in the Hertzsprung-Russell diagram differ significantly dependent on the adopted accretion rates and model assumptions. However, even before the ZAMS is reached, spectroscopic observables are unable to differentiate between these models and classical models[8].

Evolutionary calculations of accreting protostars in the literature usually fall in one of two classes. In the constant accretion scenario, the accretion rate is kept at a constant level before dropping exponentially to reach the final mass. Constant accretion rates are predicted in semi-analytic models of spherical cloud collapse, which neglect the formation of circumstellar disks[21,22]. An exponential decline occurs in the later phases because of a finite mass reservoir of collapsing prestellar clouds[23]. In the other scenario, more realistic accretion histories from hydrodynamical simulations that take the disk formation phase into account[17,20,24,25] are taken to evolve the protostar. In these models, the matter is accreted on the star through a protostellar disk, which alters notably the mass accretion history depending on the physical mechanisms of mass and angular momentum transport that operate in the disk (see methods, subsection accretion histories from hydrodynamical simulations).

Asteroseismic studies allow constraining the physics inside stellar interiors[6]. Self-driven gravity modes in γ-Doradus stars, for example, allow the determination of the core boundary mixing profile and temperature gradient from asteroseismic modelling of period spacings[26,27]. A different type of pulsator is δ-Scuti stars, which are located within the classical instability strip[28] with effective temperatures of $6500\,\mathrm{K} \le T_{\mathrm{eff}} \le 10{,}000\,\mathrm{K}$, stellar luminosities in the range $0.6 \le \log(L/L_{\odot}) \le 2$ and surface gravities in the range $3.2\,\mathrm{cm}\ \mathrm{s}^{-2} \le \log(g) \le 4.4\,\mathrm{cm}\ \mathrm{s}^{-2}$ that exist in the main sequence[29], pre-main sequence[7,8,30], and post-main sequence phase[29]. Pre-main sequence δ-Scuti stars are expected with masses from 1.5 to $3.5\,M_{\odot}$[7,8] while the upper limit for the mass of main sequence counterparts lies around $2.3\,M_{\odot}$[30]. δ-Scuti stars, independent of their respective evolutionary stage, pulsate in low order pressure modes driven by the heat engine mechanism, although turbulent pressure and especially time-dependent convection play a major role in the calculation of instability regions[8,31,32].

Here, we investigate whether the self-driven pressure modes in δ-Scuti stars can provide similar constraints as γ-Doradus stars, but for the initial model applied in stellar structure and evolution calculations. Our results show that the process of star formation leaves behind an imprint on the stellar structure that is potentially detectable in pulsation frequencies of δ-Scuti stars. This opens a window to investigate early stellar evolution in young pulsating stars that could allow many insights potentially invaluable for related fields such as exoplanet studies.

## Results
### Stellar structure of early stellar evolution
In this work, we take a $2\,M_{\odot}$ star with δ-Scuti pulsations as an example and calculate three different models of its pre-main sequence evolution using the software instrument MESA[9] with the aim to explore whether the pulsation frequencies can distinguish between the models. The first model is a classical pre-main sequence model starting from the initial assumptions described above. In addition, we calculate models in the constant accretion scenario similar to the models calculated by Steindl et al.[8]. Finally, we calculate disk-mediated models using 34 of the accretion histories from Elbakyan et al.[20], scaled to reach $2\,M_{\odot}$ after the accretion has finished.

We use the resulting equilibrium models to perform a linear non-adiabatic pulsation analysis with the stellar oscillation code GYRE[33] to obtain pulsation frequencies and growth e-folding times, the latter of which provides information on whether the pulsation mode is expected to be self-driven and, hence, observable.

Asteroseismic studies are inherently dependent on the frequencies that can be measured from observations. The expected uncertainties of frequencies obtained from time series photometry are a result of the duration of the light curve available. When studying theoretical prospects, it is common to use a conservative approach[26] to estimate the value of observational uncertainties, Δf: the Rayleigh limit,

$$\Delta f = 1/T, \qquad (1)$$

where T is the time span of the observation. To investigate whether asteroseismology can distinguish between the different models, we compare the resulting frequencies to the Rayleigh limit corresponding to state-of-the-art observations, motivated by the Kepler[4] observations in the nominal mission (4 years) and TESS[5] observations in the continuous viewing zone (357 days). This corresponds to Rayleigh limits of 0.000684 cycles per day (c/d) or about 0.0079 µHz for 4 years and 0.0028 c/d or about 0.032 µHz for 357 days according to Eq. 1.

The theoretical calculations performed in this work result in a theoretical sample of $2\,M_{\odot}$ models that range from the pre-main sequence phases until almost the terminal age main sequence, where stars have reached the end of core-hydrogen burning. Figure 1 shows the immense difference in stellar structure between the classical pre-main sequence and the disk-mediated accretion model through a Kippenhahn diagram. The classical model ignites deuterium in the centre at the age of ∼10,000 years, shown by the first appearance of a yellow burning zone in Fig. 1. After this, the convection delivers new deuterium to the centre, where nuclear burning of deuterium continues until the central deuterium abundance is depleted (i.e. after ∼100,000 years). In the disk-mediated accretion model, it takes ∼50,000 years until deuterium is first ignited. The accretion energy added to the outer parts of the stellar model results in a temperature inversion[8,19] that leads to off-centre ignition of deuterium burning. In contrast to the classical model, the position of deuterium burning inside the star is dependent on the accretion rates: the convective area retreats towards the surface during strong accretion bursts that are similar in magnitude to FU Orionis-type eruptions[34]. As the convective area retreats, deuterium added to the outer layers by accretion fails to reach the inner parts of the star. While the classical model keeps burning deuterium in the core, the disk-mediated accretion model has exhausted the central deuterium reservoir. Deuterium burning keeps producing high amounts of energy at the bottom of the convective envelope, the position of which is strongly dependent on the accretion rate. This continues until the star has reached the phase of pre-main sequence evolution, for which almost the whole star is radiative. Once, the star has reached its final mass of $2\,M_{\odot}$ and has evolved onto the

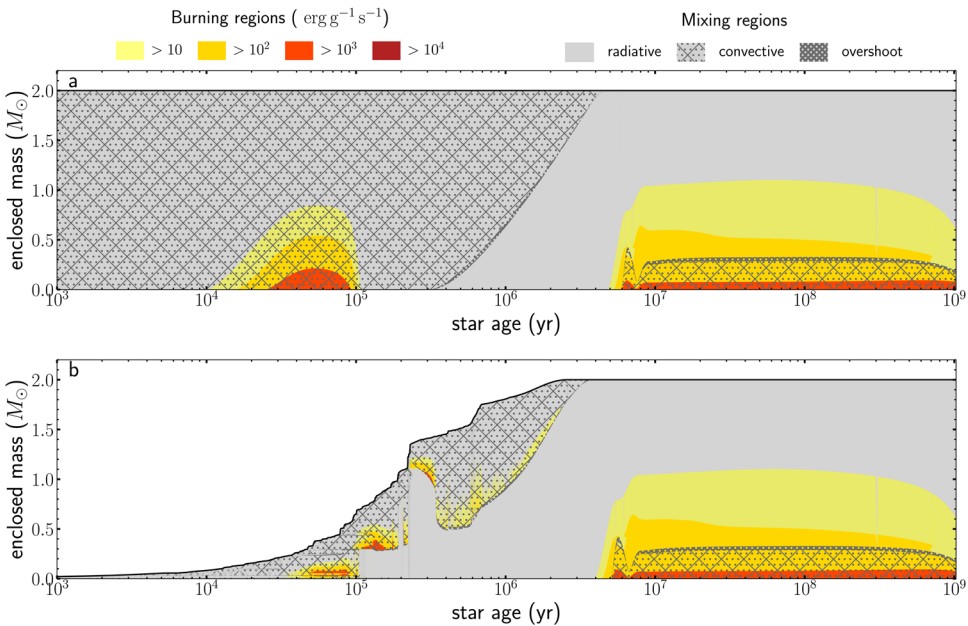

**Fig. 1 | Comparison of the internal structure of the calculated evolution models. a** Kippenhahn diagram for the classic model. **b** Kippenhahn diagram of the disk-mediated accretion model #28. In both panels, the black line shows the stellar mass as a function of the star age. Radiative parts of the stellar structure are shaded grey and different hashes mark mixing regions. The colour code from yellow to red shows nuclear burning depending on the strength. The legend applies to both panels.

ZAMS, the Kippenhahn diagrams of the classical and the disk-mediated accretion model seem identical.

Figure 2 shows the asteroseismic properties of disk-mediated model #27 in terms of the Brunt-Väisälä frequency (also referred to as the buoyancy frequency) and the Lamb frequency for $l = 1$ as an example. These frequencies are important indicators of the oscillatory structure of a star. While it is impossible to perform a direct comparison between the disk-mediated and the classical pre-main sequence models (among other things, because the disk-mediated models have different masses), we can compare snapshots of the classical model at early stages with the same stellar radius as the disk-mediated models. During the time of highest accretion (panel c of Fig. 2), the internal structure of the disk-mediated model is significantly different to the corresponding snapshot of the classical model with the same radius. While the latter is almost entirely convective, the disk-mediated model shows radiative areas in which the Brunt-Väisälä frequency is non-zero. In addition, the Lamb frequency is shifted to lower frequencies and obtains extra features in the radiative regions. But even during a quiescent phase (panel d of Fig. 2), the differences are evident: While at that moment, the classical model has obtained a radiative core that provides a non-zero Brunt-Väisälä frequency, the internal structure of the disk-mediated model still shows some characteristic patterns. Towards the end of the accretion phase (panel e of Fig. 2), the characteristic frequencies look qualitatively similar. However, changes in the internal structure still lead to changes in both the Brunt-Väisälä frequency and the Lamb frequency, especially towards the centre of the star. In the last snapshot, directly after mass accretion has finished, the characteristic frequencies of both models become similar, but small deviations in the internal structure of the models still lead to visible differences in the Brunt-Väisälä frequency.

Including the effects of accretion also significantly alters the structure and the position of the evolutionary tracks and hence the spectroscopic observables of pre-main sequence stars, shown in a Hertzsprung-Russell diagram in Fig. 3 (a Kiel diagram is available as Supplementary Fig. 1). Neither the constant accretion model nor the disk-mediated accretion model reaches the high values of stellar luminosity that is a consequence of the large radius assumed for the classical initial model.

## The imprint of star formation on stellar pulsations

We calculated pulsation frequencies at the pre-main sequence and the zero-age main sequence with the aim to compare the pulsation spectra based on the different modelling approaches (see methods, subsection stellar evolution calculations and subsection stellar pulsation frequencies). Figure 4 presents the frequency differences in these two snapshots of the evolution. During the pre-main sequence phase, the frequency differences are well above the Rayleigh limits for both the Kepler and TESS datasets. Furthermore, there is also a clear difference between the constant accretion model and the disk-mediated model. As such, the pulsation frequencies of pre-main sequence $\delta$-Scuti stars are not only sensitive to the difference between classical models and accreting models, but also to the applied accretion rate. Although the difference between the disk-mediated model and the classical model is less prominent at the zero-age main sequence, it still allows the distinction between the two models for all pulsation modes with the precision reached by data from the Kepler space mission and for a few modes with the observational precision achieved for TESS data. The spread between the constant accretion model and the disk-mediated model is larger compared to the model during the pre-main sequence phase, again pointing to the sensitivity of the pressure modes to the details of the accretion process during early stellar evolution.

The different mass accretion rates for the 34 disk-mediated accretion models (see Supplementary Fig. 2) lead to different evolutionary tracks and internal structures during the pre-main sequence phase, which manifests themselves as frequency differences (see Supplementary Figs. 3–36). Figure 5 and Supplementary Fig. 37 give an overview of the resulting changes in pulsation frequencies as a direct consequence of the accretion histories. The region in the Hertzsprung-Russell diagram occupied by the different evolutionary tracks is enormous compared to the classical pre-main sequence model or the constant accretion model.

These frequency differences essentially result from differences in the stellar radius. The radii of the disk-mediated accretion models are up to a few parts per thousand (ppt) smaller than that of the classical model at our pre-defined pre-main sequence stage. This stems from a slightly more massive inner region, as is presented in Fig. 6 and

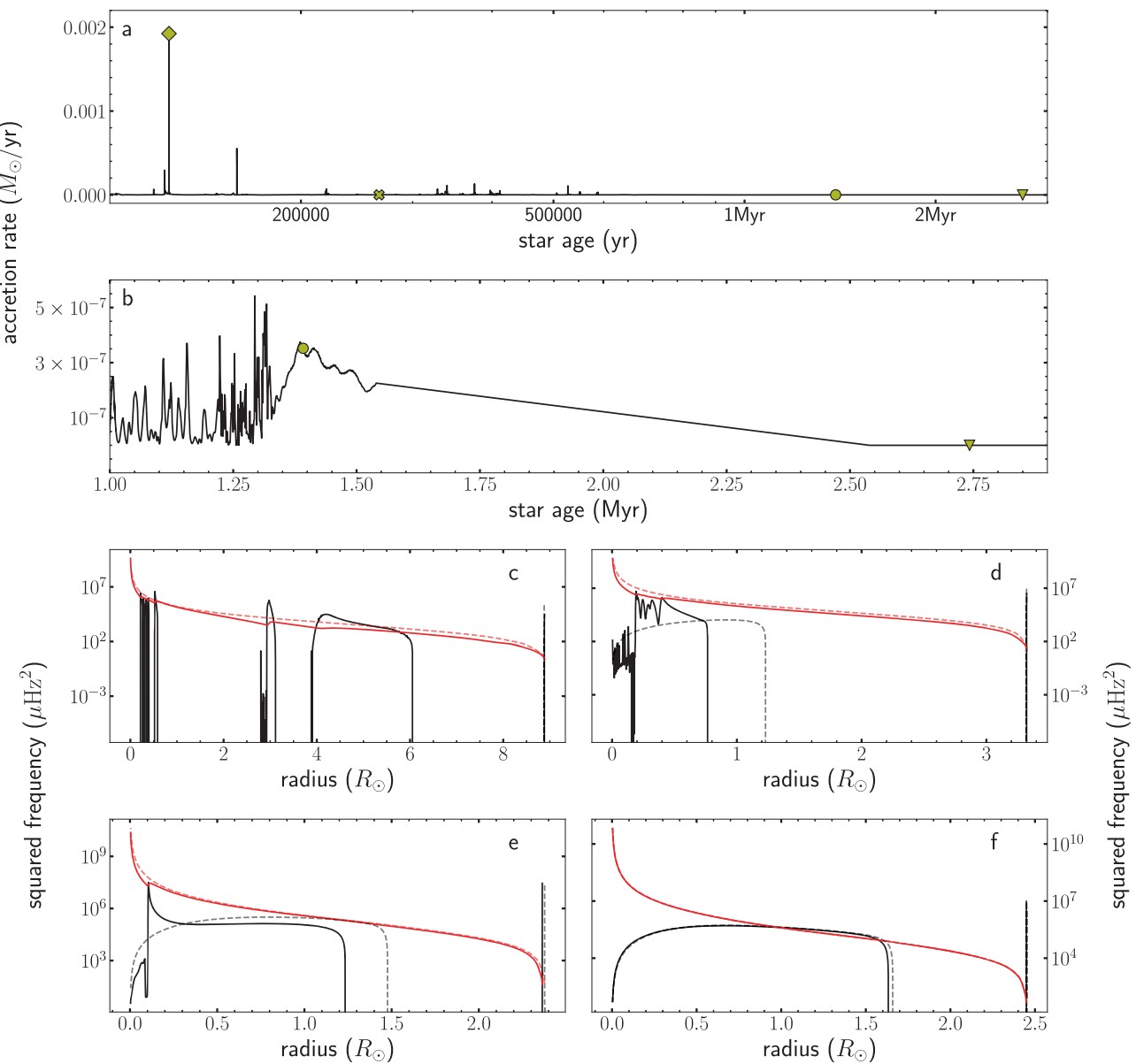

**Fig. 2 | Comparison between the asteroseismic structure of the disk-mediated and the classical evolution model at early stages. a** Mass accretion history of model #27. **b** Zoom into the mass accretion history of model #27 towards the end of the accretion phase. Yellow symbols mark specific positions. **c–f** Squared Brunt-Väisälä frequency (black) and squared Lamb frequency for $l = 1$ (red). The solid line corresponds to the disk-mediated model while the dashed line corresponds to a snapshot of the classical pre-main sequence model with the same radius for comparison. **c** The model during a high accretion phase (diamond). The classical model in this case is almost entirely convective, hence, the squared Brunt-Väisälä frequency is negative throughout the star other than just below the stellar surface. **d** The model during a quiescent phase (cross). **e** The model towards the end of the accretion phase (circle). The classical model is shown at the moment of the smallest radius, since the classical model never reaches this small radius in the early stages. **f** The model soon after the accretion has finished, and the star has obtained its final mass (triangle).

Supplementary Fig. 38. Pressure mode pulsations are sensitive enough to feel this size difference and translate them into frequency differences. Once the models are at our pre-defined stage of the ZAMS, the sizes are on average equal, but there is still a spread in both stellar radii and frequency difference comparable to the Rayleigh limit for TESS data. This spread of pulsation frequencies within the 34 different accretion models provides an additional uncertainty in theoretically calculated pressure mode pulsation frequencies for $2\,M_\odot$ δ-Scuti stars only from the (unknown) accretion past.

Considering every excited mode for all 34 of the $2\,M_\odot$ δ-Scuti star models with our standard input physics, we find that currently available space photometry provides sufficient precision to distinguish between classical and more realistic pre-main sequence models.

During the pre-main sequence phase, the differences in pulsation frequencies are larger than Kepler precision for all excited modes in our models, while they are larger than TESS precision for about 73% of all excited modes (see Table 1).

The mixing of chemical elements in the radiative envelope plays an important role in stellar evolution, feeding the convective core with additional fuel and thus changing the evolutionary path[27]. To investigate the dependence of our findings on the choices of this input physics, we repeated our calculations with different values for the amount of envelope mixing and overshooting. The additional mixing in the stellar envelope results in a larger radius and frequency differences during the pre-main sequence stage of evolution (see Table 1, Fig. 7, Supplementary Figs. 39 and 40 and methods, subsection stellar

evolution calculations) as compared to our standard input physics. As all excited modes show frequency differences larger than the precision yielded by Kepler and TESS observations, it is evident that the improved modelling of the early stellar evolution results in changes which are detectable by means of pulsations in pre-main sequence $\delta$-Scuti stars. The calculations with more overshooting lead to smaller differences during the pre-main sequence and at the ZAMS (see Supplementary Figs. 41 and 42), but the frequency differences remain large enough to be observable with Kepler precision.

Pre-main sequence $\delta$-Scuti stars become unstable to pulsation even before the evolutionary stage at which we have presented our

results above[8]. To compare the different models at earlier evolutionary stages, we additionally calculated the frequency spectra of models with a fixed radius of $2.8\,R_\odot$, which the stars obtain at an effective temperature of around 7300 K. While this does not present an equal evolutionary stage, it serves as a guide to investigating differences in stellar structure. As expected, the frequency differences between the classical and the disk-mediated pre-main sequence models for some radial orders are even larger at this earlier evolutionary stage (see Table 1 and Fig. A39).

## Discussion

Our results are in strong contrast with the common belief that the physical details of early stellar evolution (i.e. star formation, mass accretion process, etc) have no influence on the stellar structure and suggest that the accretion history should be included in our asteroseismic interpretation of pre-main sequence $\delta$-Scuti stars. Although the frequency differences are larger than the expected precision of state-of-the-art observations, we find that the effects described in this work are of lesser extent compared to other input parameters in stellar structure (stellar mass, metallicity, convective overshooting, etc.) and thus can only be applied once a sufficiently constrained model has been found already. As such, this manuscript provides a similar analysis to Michielsen et al.[26], which constitutes the basis for many impressive recent results regarding the mixing profile of gravity mode pulsators[27]. In a similar fashion, this work demonstrates that forward asteroseismic modelling of pre-main sequence $\delta$-Scuti stars should provide an opportunity to test the imprint of star formation on their pulsation frequencies (see methods, subsection stellar evolution calculations).

Zwintz et al.[7] showed that multiple well-known pre-main sequence stars would be in the perfect evolutionary stage to launch an in-depth study and perform asteroseismic modelling with disk-mediated pre-main sequence models. Unfortunately, young open clusters have been avoided by the main Kepler mission, and the TESS continuous viewing zones point far off the galactic plane. The latter is the region of the sky where most young open clusters are located. Hence, we are currently

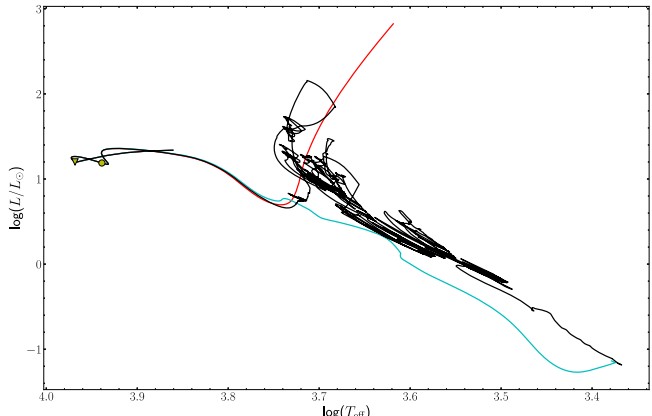

**Fig. 3 | Comparison between classical, constant accreting and a selected disk-mediated model in the Hertzsprung Russel diagram.** The red line shows the classical model, the turquoise line shows the constant accretion model and the black line shows the disk-mediated accretion model. The yellow circle marks the pre-defined pre-main sequence (i.e. central carbon mass fraction drops to a value of $10^{-4}$) while the yellow triangle marks the ZAMS (i.e. central hydrogen mass fraction has dropped by 0.01 in comparison to the initial value). All evolutionary tracks shown here have been calculated with the standard input physics.

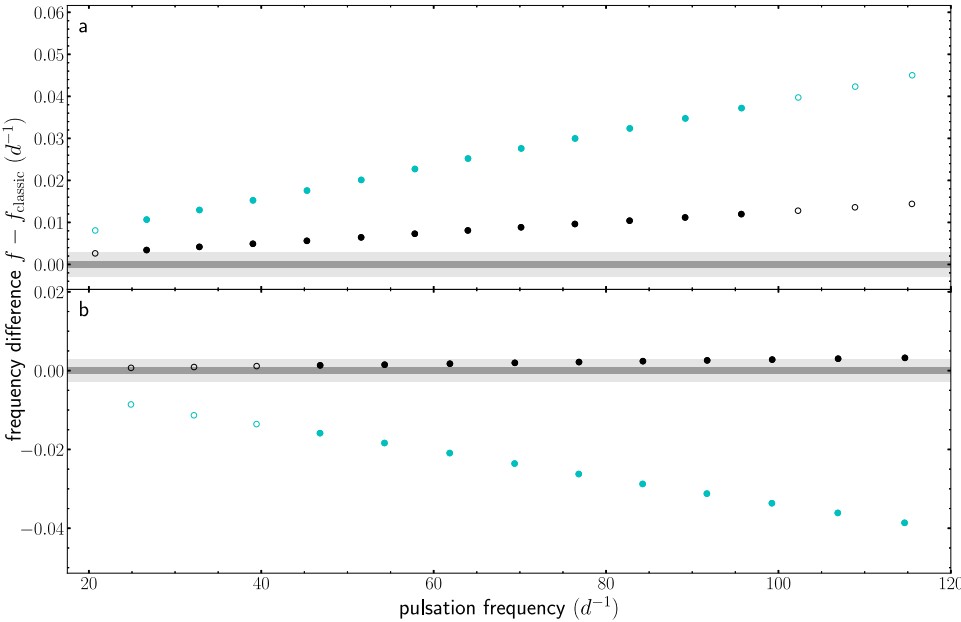

**Fig. 4 | Resulting frequency differences for different evolutionary calculations.** The frequency difference of $l = 1, m = 1$ modes as a function of the pulsation frequency. The black (turquoise) circles correspond to differences between the disk-mediated accretion (constant accretion) model and the classical model. Unstable modes are shown as filled circles, while stable modes are depicted as open circles. The grey areas mark the Rayleigh limit corresponding to 4 year Kepler (dark grey) and 357 days TESS light curves (light grey). **a** Results at our pre-defined pre-main sequence (i.e. central carbon mass fraction drops to a value of $10^{-4}$). **b** Results at the ZAMS stage (i.e. central hydrogen mass fraction has dropped by 0.01 in comparison to the initial value).

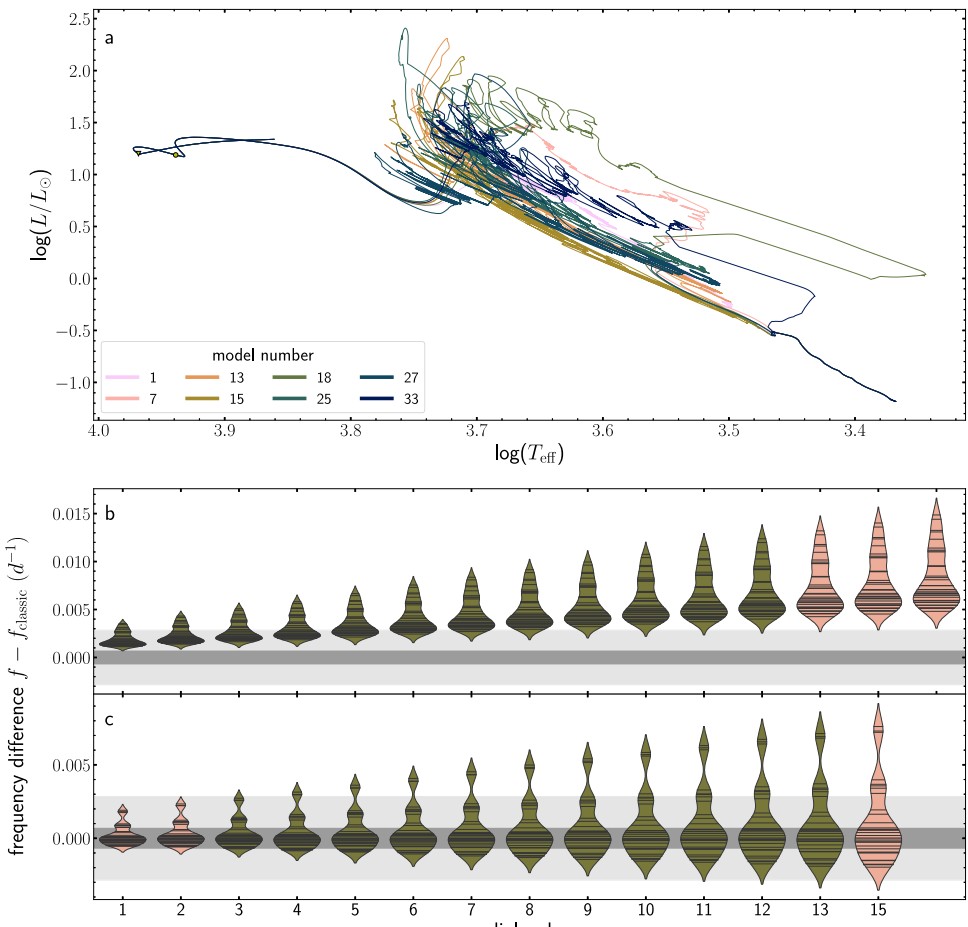

**Fig. 5 | Comparison between the different disk-mediated accretion histories for the standard input physics. a** Evolution in a Hertzsprung-Russell diagram of eight chosen models. These models have been selected to show the extent in the Hertzsprung-Russell diagram reached by different accretion histories while minimising the overlap between lines. See Supplementary Fig. 37 for a version with all models. **b** The distribution of frequency values in the 34 models for the pre-main sequence phase. The frequency differences are presented as violin plots grouped according to the radial order. In this type of plot, each horizontal line corresponds to the value of one model and a kernel density estimate calculated with a bandwidth of 0.3 on each side leads to a violin-like shape. Violins are plotted green if the modes are excited in most of the models or salmon if they are stable. The grey areas mark the Rayleigh limit corresponding to 4 year Kepler (dark grey) and 357 days TESS light curves (light grey). The colours of the evolutionary tracks in the top panel are chosen without specific reasoning and are only to allow discrimination between the models. **c** Distribution of frequency values at the ZAMS.

unable to perform such a study due to the lack of suitable observations. Predictions from this work can only be verified observationally once long time series photometry from space of pre-main sequence $\delta$-Scuti stars become available. We remain hopeful that the next prime mission for such observations, ESA's PLATO mission, will provide observations that close this gap. Besides that, the ideal solution for this problem could be a dedicated small-scale space mission, providing long baseline observations of young open clusters. Once applicable observations are available, we propose analysis according to the statistical procedure of forwarding asteroseismic modelling described by Aerts et al.[35], which predominantly relies on the observed frequency values. Classical constraints (such as $T_{\mathrm{eff}}$, $\log(g)$, $[\mathrm{M/H}]$, and $\log(L/L_\odot)$) are of significant importance to limit the parameter space of initial calculations as well as to eliminate models a posteriori (see methods, subsection forward asteroseismic modelling of pre-main sequence $\delta$-Scuti stars). Observational proof of the potentially detectable signatures of the imprint on star formation of stellar pulsations hence also relies on successful mode identification and accurate spectroscopic parameters (for example, ±150 K in $T_{\mathrm{eff}}$ and ±0.1 dex in $\log(g)$.

The issue of unrealistic initial models is not unique to $\delta$-Scuti stars but applies to all types of stars across the observable mass range.

Hence, a similar investigation of, for example, higher mass stars that are expected to reach the main sequence long before ending their respective evolution as accreting objects will be important for progress, in particular for asteroseismology of $\beta$-Cephei or Slowly Pulsating B stars.

## Methods

### Accretion Histories from hydrodynamical simulations

The mass accretion rates were taken from Elbakyan et al.[20] based on the numerical hydrodynamics model of Vorobyov & Basu[25]. This model computes the formation and long-term evolution of circumstellar disks starting from the gravitational collapse of rotating prestellar clouds, taking disk self-gravity, stellar irradiation heating, disk radiative cooling and turbulence into account. The effects of turbulence were considered using the $\alpha$-model of Shakura & Sunyaev[36] with a constant $\alpha$-value set equal to 0.005. The use of the thin-disk limit allows computing the disk dynamical evolution for up to 1–2 Myr, which is not achievable with full 3D models. The mass accretion rates were computed as the mass of gas passing per unit time through the inner sink cell. The radius of the sink cell was set equal to 5 au to reduce the stringent requirements on the hydrodynamic timestep imposed by the Courant-Friedrichs-Levy condition. The accretion

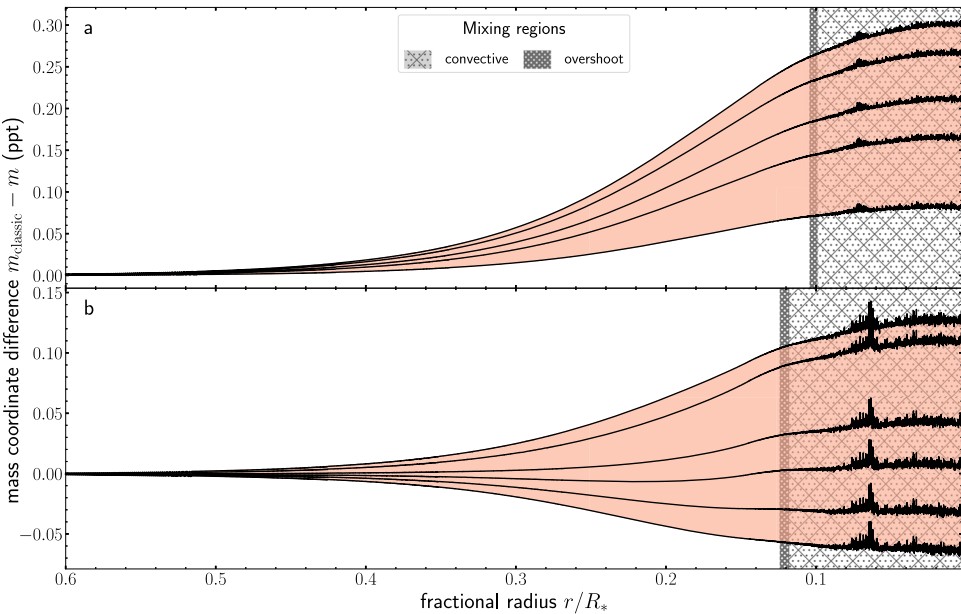

**Fig. 6 | Differences in stellar structure between the disk-mediated accretion models and the classical model.** The mass coordinate as a function of the fractional stellar radius. The grey shaded areas mark the extension of the convective core and the overshooting region according to the legend. The salmon shaded area marks the range of mass coordinate differences reached by the 34 disk-mediated models. **a** Differences in structure at the pre-main sequence stage. Selected models shown are from bottom to top: 31, 33, 2, 34, 13, 6. **b** Differences in structure at the ZAMS. Selected models shown are from bottom to top: 30, 7, 3, 12, 16, 25. The models in panels **a** and **b** have been selected to show the boundaries of the shaded area and to minimise the overlap within the lines. See Supplementary Fig. 38 for a version with all models.

**Table 1 | Overview of the different parameters for the stellar evolution calculations used in this work and summary of potentially detectable frequency differences**

|  | Standard input physics | Enhanced envelope mixing | Enhanced overshooting |
|---|---|---|---|
| $D_{min}$ | 1 cm² s⁻¹ | 5 cm² s⁻¹ | 1 cm² s⁻¹ |
| $f$ (top) | 0.01 | 0.01 | 0.05 |
| $f_O$ (top) | 0.005 | 0.005 | 0.025 |
| $f$ (bottom) | 0.005 | 0.005 | 0.025 |
| $f_O$ (bottom) | 0.0025 | 0.0025 | 0.0125 |
| **Pre-main sequence** |  |  |  |
| Probing power Kepler | 100% | 100% | 63% |
| Probing power TESS | 73% | 100% | 0% |
| **ZAMS** |  |  |  |
| Probing power Kepler | 47% | 100% | 28% |
| Probing power TESS | 9% | 98% | 0% |
| **Pre-main sequence** (radius = 2.8 $R_\odot$, see methods, subsection stellar evolution models) |  |  |  |
| Probing power Kepler | 87% | — | — |
| Probing power TESS | 45% | — | — |

This table gives the different parameters for the input physics chosen in this work (see methods, subsection stellar evolution models). Additionally, we provide the percentages of excited pulsation modes that have probing power for Kepler and TESS precision. A pulsation mode is expected to have probing power if the frequency differences between the classical and the disk-mediated models is larger than the Rayleigh limit for observations. These values are listed for the three distinct evolutionary stages presented in this work (see methods, subsection stellar evolution models). The pre-main sequence model at a radius of 2.8 $R_\odot$ has only been calculated for the standard input physics.

rates in the late disk evolution phase (after 1–2 Myr) were manually tapered off to zero to take the disk photoevaporation into account[37]. The model reproduces the observed slope of the mass accretion rate vs. stellar mass dependence[38,39] in young star-forming regions and also can explain the luminosity problem for young low-mass stars[40]. We note that the accretion rates derived with the model of Vorobyov & Basu[38,39] are qualitatively similar to those of Bae et al.[41] and Vorobyov et al.[42], who used a smaller sink cell and the layered model of magnetised disks of Armitage et al.[43]. The main feature of all these models is that the matter is transported from the prestellar cloud to the forming star through the circumstellar disk, unlike the spherical collapse models[22] in which the disk phase is absent. As a result, the mass accretion rates are disk-mediated and can exhibit complex behaviour such as accretion bursts similar to those observed in FU Orionis and EX Lupi-type objects[34].

For the initial conditions, Elbakyan et al.[20] have chosen a span of initial cloud masses from 0.061 to 1.69 $M_\odot$ and initial ratios of rotational to gravitational energy from 0.25 to 11.85%. These numbers lie within the observationally inferred values for prestellar clouds[44]. The resulting accretion histories lead to stellar masses between 0.0468 and 1.319 $M_\odot$. All mass accretion rates include an early part with high mass accretion rates (corresponding to the pre-disk stage), an abrupt drop to no accretion (manifesting the formation of a centrifugally balanced disk), and subsequent continuation of mass accretion rates that gradually decline with time while experiencing bursts up to $10^{-3} M_\odot$/yr in several models. The calculation of stellar structure models with these mass accretion rates sometimes experiences convergence problems in the first few thousand years. To withstand this issue, we use the beginning of #28 for every mass accretion history, the consequence of which is that the mass accretion rate until the first drop to zero is the same for every model calculated in this work. During the remaining accretion time, the accretion rate is scaled up to ensure that the final model obtains 2 $M_\odot$. Given the different final masses in the original mass accretion rates, this scaling factor is different for every calculation. Original mass accretion rates that lead to very low-mass stars are

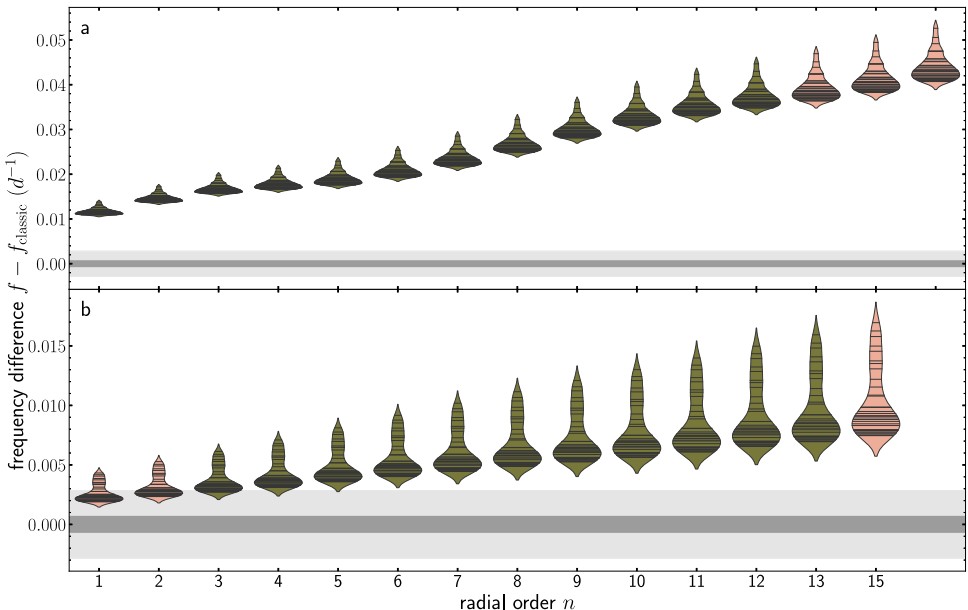

**Fig. 7 | Differences in pulsation frequency between the 34 disk-mediated accretion models and the classical models for the input physics with increased envelope mixing.** The violin plots are similar to those shown in the lower panels in Fig. 5. The grey areas mark the Rayleigh limit corresponding to 4 year Kepler (dark grey) and 357 days TESS light curves (light grey). **a** Results at the pre-main sequence stage. **b** Results at the ZAMS.

subject to larger scaling factors as a consequence. For the lowest final mass, model #17, the scaling results in mass accretion being too high at the early stages for the MESA model to converge and is hence omitted from this study. All adjusted mass accretion rates are shown in Supplementary Fig. 2. The scaling procedure assumes that the mass accretion rate histories for stars of $2\,M_\odot$ do not differ qualitatively from their lower-mass counterparts, which we expect to be the case. An increase in the stellar mass results in a more stable disk against gravitational instability, but this effect can be counterbalanced by a higher mass infall rate on the disk[24] in more massive initial clouds. Furthermore, similar but scaled-up mass accretion rates, including accretion bursts, were recently obtained in 3D numerical gravito-radiation hydrodynamics simulations of massive protostars[45], implying a qualitative similarity in the mass accretion histories across the stellar mass spectrum.

## Stellar evolution models

Stellar evolution calculations in this work have been performed with the software instrument Modules for Experiments in Stellar Astrophysics, MESA[9,46–49] version v-12778. MESA is a Henyey style code with a coupled solution for the structure and composition equations[9]. This work hence relies on the calculation of one-dimensional spherically symmetric stellar models for which we use the standard MESA equation of state[9] and the OPAL opacity tables[50]. Our calculations of the equilibrium stellar model ignore rotation and magnetic fields. While stellar rotation is an important ingredient in stellar evolution and stellar pulsations, computational asteroseismology is currently not able to include these effects in the calculation of the equilibrium model since the physical descriptions are missing. Hence, we apply the standard approach in asteroseismic studies[26,27] and apply the effect of stellar rotation on the pulsation frequencies during the pulsation analysis of the equilibrium model with GYRE.

The mixing of chemical elements by convection is treated in MESA as a diffusive process governed by a diffusion coefficient $D$. In convective areas, the diffusion coefficient $D_{\mathrm{conv},0}$ is calculated via the mixing length description of Cox & Giuli[51] with a mixing length of $\alpha_{\mathrm{MLT}} = 2.2$ and the Ledoux criterion as well as the description of convective premixing to find convective boundaries[48]. Beyond these

boundaries, we employ exponential overshooting according to

$$D_{\mathrm{OV}} = D_{\mathrm{conv},0}\exp\left(\frac{-2z}{fH_p}\right), \qquad (2)$$

where $z$ is the distance to the radiative layer and $H_p$ is the pressure scale height. An additional parameter $f_0$ describes the distance $f_0H_p$ from the convective boundary into the convective zone at which the switch to overshooting occurs. The parameters for overshooting are set to $f = 0.01$ and $f_0 = 0.005$ at the top of any convective zone and to $f = 0.005$ and $f_0 = 0.0025$ at the bottom. The minimum mixing coefficient throughout the star is set to $D_{\min} = 1\,\mathrm{cm}^2\mathrm{s}^{-1}$ for our standard input physics. We use a uniform initial composition of $X = 0.734$, $Y = 0.252$ and an initial metallicity of $Z = 0.014$. Relative abundances for the chemical elements beyond Helium are set according to the solar composition presented in Asplund et al.[52], with additional updates to some key elements based on Nieva & Przybilla[53] and Przybilla et al.[54]. This set-up is one of the pre-defined solar compositions in MESA. The updated abundances in the form $\mathrm{eps\_EL} = \log_{10}(\mathrm{EL/H}) + 12$, where EL is a placeholder for elements, are: eps_C = 8.33, eps_N = 7.79, eps_O = 8.76, eps_Ne = 8.09, eps_Mg = 7.56, eps_Al = 6.30, eps_Si = 7.50, eps_S = 7.14, eps_Ar = 6.50 and eps_Fe = 7.52. The composition of the models calculated in this work includes 20 parts per million (ppm) of $^2$H and 85 ppm of $^3$He. In terms of atmospheric boundary conditions, we found that the temperature-opacity (T-$\tau$) relation by Eddington[55] successfully recreates the instability region for pre-main sequence $\delta$-Scuti stars[8]. Hence, we use the Eddington atmosphere for our calculations in this work.

We build our initial model by creating a pre-main sequence model with a mass of $0.04\,M_\odot$ according to the standard MESA routing. After that, $0.03\,M_\odot$ are removed by the standard mass-relax scheme. We then follow the contraction of the remaining $0.01\,M_\odot$ protostellar seed until its radius reaches $1.5\,R_\odot$. This procedure is identical to the one used in previous studies[18] and leads to initial models that represent an evolved second Larson core with high entropy value[18,56]. To model the accretion onto a one-dimensional stellar model, we follow a description applied in the literature[16] in the view of a non-spherical accretion process[14,15] that allows the star to radiate its energy

over most of the photosphere. The energy budget of accretion is governed by gravitational energy per unit of accreted mass $-GM/R$ and the internal energy per unit mass of the accreted material $+\epsilon GM/R$, where $G$ is the gravitational constant, $M$ and $R$ are the mass and radius of the accreting star and the value $0 \le \epsilon \le 1$ describes the geometry of accretion[15,57]. In this study, we use $\epsilon = 0.5$, corresponding to accretion from a thin disk around the equator also applied in the previous studies[15,19,57]. The total energy budget is therefore

$$\frac{dE_{acc}}{dt} = (\epsilon - 1)\frac{GM\dot{M}}{R}, \tag{3}$$

where $\dot{M}$ is the mass accretion rate. A second parameter, $\beta$, then controls how much of the net energy is absorbed by the stellar envelope ($L_{add} = \beta\epsilon\frac{GM\dot{M}}{R}$) and how much is radiated away as accretion luminosity ($L_{acc} = (1-\beta)\epsilon\frac{GM\dot{M}}{R}$). For $\beta = 0$, no energy is absorbed by the star and the process is referred to as cold accretion, while the case of $0 < \beta \le 1$ is referred to as hot accretion. $\beta$ is a free parameter that is expected to vary with the time-dependent-accretion rate[17,19,20,56]. We choose a constant value of $\beta = 0.1$ for the constant accretion scenario and functional dependence on the mass accretion rate previously applied in the literature[19] for the disk-mediated accretion models. The latter is given by a step function with a smooth transition:

$$\beta(\dot{M}) = \frac{\beta_L \exp\left(\frac{\dot{M}_m}{\Delta}\right) + \beta_U \exp\left(\frac{\dot{M}}{\Delta}\right)}{\exp\left(\frac{\dot{M}_m}{\Delta}\right) + \exp\left(\frac{\dot{M}}{\Delta}\right)}. \tag{4}$$

Here, $\beta_L = 0.005$ and $\beta_U = 0.2$ are the lower and upper bound of $\beta$ and $\dot{M}_m = 6.2 \times 10^{-6} M_\odot/\text{yr}$ and $\Delta = 5.95 \times 10^{-6} M_\odot/\text{yr}$ are the midpoint and the width of the crossover between the lower and upper limit.

The energy absorbed by the star is added as extra heat to the MESA models. The distribution of extra heat within the star is an additional free input choice for the model. Applied methodologies in the literature include uniform distribution[57] which is most likely non-physical, a step function in the outer layers of the star[19], and a linear increase as a function of the mass coordinate[18]. We follow the last approach as we believe this to be the most physical of the three, such that we add the extra heat according to

$$l = \frac{L_{add}}{M} \max\left\{0, \frac{2}{M_{outer}^2}\left(\frac{m_r}{M} - 1 + M_{outer}\right)\right\}. \tag{5}$$

Here, $M_{outer}$ defines the fractional mass of an outer region in which the extra heat $l$ is deposited and $m_r$ corresponds to the mass coordinate. We choose a value of $M_{outer} = 0.01$ for the disk-mediated accretion models, meaning that the accretion energy absorbed by the star is entirely deposited within the outer 1% of the stellar model. For the constant accretion model, we follow the approach of Steindl et al.[8] and choose $M_{outer} = 0.1$. The composition of the accreted material is assumed to be invariable with time and corresponds to the initial composition of the star.

We save the stellar model at specific points of the evolution for the subsequent calculation of pulsation mode frequencies and analysis. These points are pre-defined by the central composition of carbon (pre-main sequence) and hydrogen (main sequence). The temperature inversion resulting from the accretion process makes a comparison between the accreting models and the classical model unreasonable during the early phases of evolution. Only after this temperature inversion has vanished, we can safely compare the two evolutionary calculations. We define one point on the pre-main sequence, where the central carbon mass fraction drops to a value of $10^{-4}$. This corresponds to the phase of the first CNO ignition: the stellar core is hot enough to burn carbon to nitrogen but is not yet able to follow through with the remainder of the CNO cycle. This is an evolutionary stage in which pre-

main sequence $\delta$-Scuti stars can be observed typically[7,8,58]. After the initial abundance of carbon is depleted, the stellar core again heats up until the full CNO cycle can operate in equilibrium - the main sequence phase of evolution. In the lack of a better definition of the ZAMS, we define the ZAMS where the central hydrogen abundance, $X_c$, has dropped by 0.01 compared to the initial value. At this point, however, the star is, in fact, already on the main sequence for a short period of time. In order to verify that the origin of the difference in stellar pulsation frequencies does not lie in slightly different evolutionary stages, we ensured that the difference in the hydrogen mass fraction is smaller than $2 \times 10^{-6}$ and the difference in the carbon mass fraction is smaller than $2 \times 10^{-9}$. Furthermore, we also checked that the change in pulsation frequency for one timestep is smaller than Kepler precision.

The individual models are presented at two specific evolutionary stages in the main text: (1) The pre-main sequence, where the central carbon mass fraction drops to a value of $10^{-4}$ due to the first onset of the CNO cycle. (2) The zero-age main sequence, for this purpose defined as the point when the central hydrogen mass fraction has dropped by 0.01 compared to the initial value: $X_c = X_{c,\text{init}} - 0.01$.

When calculating the stellar evolution model, we tried to minimise the influence of numerical errors as much as possible. This includes using higher than standard spatial and temporal resolution throughout the stellar model, where we also additionally increased the spatial resolution around specific regions such as the overshooting region. In addition, we performed various simple tests, to investigate the set-up with regard to numerical stability when applying small changes. As such, we can confirm that the results are stable against small changes in the applied accretion factor or the temporal and spatial resolution. Additionally, the set-up produces the same results when saving a MESA model when the accretion has finished and reloading it within an identical copy of the inlists we used to calculate the classical evolution model.

The main result of this work is that the pulsation frequencies of evolutionary calculations, including the mass accretion phase, are significantly different from the pulsation frequencies of the classical evolution model. We verified that this is the case independent of the choice of input physics for chemical mixing by changing one parameter at a time. We perform calculations with five times the envelope mixing ($D_{min} = 5\,\text{cm}^2\text{s}^{-1}$) as well as five times the convective overshooting ($f = 0.05$ and $f_0 = 0.025$ at the top and with $f = 0.025$ and $f_0 = 0.0125$ at the bottom). The corresponding results are presented in Table 1, Fig. 7 and Supplementary Figs. 39, 40, 41 and 42. The differences in pulsation frequencies between the accreting models and the classical models change, dependent on the input physics chosen, but the discrepancy remains. All different sets of models include pulsation modes with differences larger than the expected uncertainty for observed frequencies, which verifies the main result as stated above. We furthermore note that the frequency differences are larger for models calculated with a higher amount of envelope mixing. The latter is a promising result since the value of envelope mixing chosen in the standard input physics for this work represents very low values, compared to i.e. the SPB star KIC 8324482 for which asteroseismic modelling leads to an envelope mixing coefficient of $\log D_{min} = 3.125^{+0.125}_{-0.250}\,\log(\text{cm}^2\text{s}^{-1})$[59].

We additionally present results at an earlier evolutionary stage in the calculations where the stellar radius reaches $2.8\,R_\odot$ after entering the instability region for pre-main sequence $\delta$-Scuti stars[8]. Because the different evolutionary pathways lead to different stellar radii throughout the evolution, this does not present an independent measure of the evolutionary stage. Nevertheless, it allows us to present the differences between the classical and disk-mediated models further away from the ZAMS. For the calculations, similar to the case of central carbon or hydrogen abundance, we ensured that the difference in radius is smaller than $1 \times 10^{-8}\,R_\odot$. Supplementary Figs. 43 and 44 present the resulting frequency differences for this case. The

discrepancy between the classical and disk-mediated models is even larger compared to the stages described by the central carbon and hydrogen abundance. The larger imprint from the accretion process onto the stellar structure is expected, as less time has passed since the former occurred. The spread between the frequencies of the disk-mediated models themselves is also much larger, reaching more than 0.3 c/d for the fundamental dipole mode.

## Stellar pulsation frequencies

The equilibrium models resulting from the MESA calculations are used as input for the subsequent calculation of non-adiabatic theoretical pulsation frequencies with version 6.0.1 of the stellar oscillation code GYRE[33,60,61]. Pulsations of $\delta$-Scuti stars are most commonly observed in radial and dipole modes[62] as higher order modes suffer from cancellation effects[29]. In this work, we present results for dipole modes (harmonic degree $l = 1$, azimuthal order $m = 1$), but the results for radial modes are virtually identical. We calculate pulsation modes with radial orders $0 < n \leq 50$ in the frequency range $5 \leq f \leq 150$ c/d. GYRE calculates the oscillation frequencies as solutions to the equations of linear stellar oscillations and assumes a time dependence proportional to $\exp(-i\sigma t)$, where $t$ is the time, $i$ is the imaginary unit and $\sigma = \sigma_R + i\sigma_I$ the complex eigenfrequencies. Hence, the period of oscillation is given by $\Pi = \frac{2\pi}{\sigma_R}$ and the imaginary part of the eigenfrequency gives rise to the growth e-folding time of the pulsation amplitude $\tau = \frac{1}{\sigma_I}$. For positive growth e-folding times, we expect the pulsation mode to be excited, while negative growth e-folding times correspond to stable modes. The effect of rotation is included by means of a first-order perturbative approach known as Ledoux splitting[26,29]. For the calculation of the perturbation, we assume uniform rotation at 20% of the critical rotation velocity. At this rotational velocity, the centrifugal acceleration and, hence, the flattening of the star can be ignored[26]. Furthermore, the rotation frequency is much smaller than the pulsation frequencies, rendering such a first-order perturbative approach adequate. This computational set-up is certainly not adequate for fast rotating stars, but many pre-main sequence $\delta$-Scuti stars are found to be moderately or slowly rotating stars[7] for which the prescription of rotation used in this work is adequate.

## Forward asteroseismic modelling of pre-main sequence $\delta$-Scuti stars

Once observations of a suitable pre-main sequence $\delta$-Scuti star are available, forward asteroseismic modelling, according to Aerts et al.[35] can be undertaken to investigate which types of pre-main sequence models (disk-mediated, constant accretion scenario, classical) provide the best accordance with the observed pulsation frequencies. The methodology has been developed for gravity mode pulsations and has recently been applied to constrain the internal mixing of such stars[27], but the statistical techniques presented in Aerts et al.[35] are applicable to different types of pulsators as well. The input needed to perform asteroseismic modelling includes, next to classical constraints such as $T_{eff}$, $\log(g)$, $[M/H]$, and $\log(L/L_\odot)$[27], additional mode identification. For pre-main sequence $\delta$-Scuti stars, mode identification can be obtained from regularities in echelle diagrams, as has been shown in the past[63–65].

In the methodology described by Aerts et al.[35], the calculation of the merit function, preferably the Mahalanobis distance, does not include the classical constraints. In contrast, these are used a posteriori to evaluate the goodness of the model[35]. In applications to gravity mode pulsators, they are typically used to determine the range of initial parameters for the calculation of a model grid or to a posteriori eliminate stellar models outside of the observed regime[27,66,67].

Nevertheless, such modelling work would be an immense computational effort as thousands of evolutionary calculations would have to be calculated. Disk-mediated mass accretion rates pose a challenge to the stellar evolution code MESA. As a consequence, calculations of such evolutionary models typically take at least 100 hours of execution time (on 10 cores of Intel Xeon X5650 processors). In this regard, accurate knowledge of the classical constraints $T_{eff}$, $\log(g)$, $[M/H]$, and $\log(L/L_\odot)$ would mean an immense reduction of the computational burden. The prospect of constraining the early phases of star formation, however, would make such an endeavour worthwhile.

## Data availability

The data that support the findings of this study are publicly available in zenodo with the identifier https://doi.org/10.5281/zenodo.6762319[68].

## Code availability

The stellar evolution code, MESA, is freely available and documented at https://docs.mesastar.org/en/release-r22.05.1/. The stellar pulsation code, GYRE, is freely available and documented at https://gyre.readthedocs.io/en/stable/. The input files of MESA and GYRE to reproduce the results of this study are available from the corresponding author upon reasonable request. The scripts to produce all Figures including supplementary material have been deposited in a zenodo repository with the identifier https://doi.org/10.5281/zenodo.6762319[68].

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

## Acknowledgements
We are grateful to the developers of MESA and GYRE for providing and maintaining the publicly available stellar structure and stellar pulsation code. We acknowledge funding from the University of Innsbruck (to T.S. and K.Z.) and Ministry of Science and Higher Education of the Russian Federation under the grant 075-15-2020-780, N13.1902.21.0039, section: Methods (to E.V.)

## Author contributions
T.S. wrote code to include the protostellar accretion phase in MESA, implemented and applied the modelling procedures, interpreted the results and wrote part of the text. K.Z. defined the research, interpreted the results and wrote part of the text. E.V. provided the disk-mediated mass accretion rates and wrote part of the text. All authors contributed to the discussions and have read and iterated upon the text of the final manuscript.

## Competing interests
The authors declare no competing interests.
