## [Peer Review File · Nature Communications]

REVIEWER COMMENTS

Reviewer #1 (Remarks to the Author):

I thank the authors for taking, to a large extent, my comments on the previous version of the manuscript into account, in the resubmission. However, my main concern as to the suitability of the manuscript for a journal aimed for a broad readership remains: the manuscript in its title and abstract emphasizes the possibility of asteroseismic investigation of the detailed effects of the early evolution of stars, yet the claimed detectability relies on minute changes in the stellar radius at fixed evolutionary stages, characterized by the central abundances of carbon or hydrogen; I see no way that these points can be identified independently by other observations, allowing to detect the effect of the treatment of accretion. This concern is not addressed by the 'rebuttal' by the authors to the earlier referees' reports.

Concerning the results in the main text, as illustrated in Fig. 4, I note that only in the PMS model is the frequency shift significant, when the full set of models is considered. For the ZAMS model, the effect is essentially solely an increase in the theoretical uncertainty of the frequencies, with a slight bias.

As before, I do find the results interesting from the point of view of stellar evolution and hence, with suitable rewriting, for a specialized journal. Here it would also be interesting to relate the behaviour of the convective envelope, illustrated in Fig. 1 but perhaps analysed for a 1 Msun case, to the issues of differential composition differences, as discussed by Melendez et al. (2009; <https://ui.adsabs.harvard.edu/abs/2009ApJ...704L..66M/abstract>) and the preprint by Nordlund (<https://ui.adsabs.harvard.edu/abs/2009arXiv0908.3479N/abstract>). In support of such a revision, I have some additional comments:

- p. 3, l. 122 - 123: this is somewhat confusing: one would surely not describe a growth e-folding time as a growth rate. The growth rate is what is determined as the imaginary part of the frequency, from which the growth time (or, if you like, the growth e-folding time) can be calculated.

- p. 6, l. 240 - 241, Fig. 4: I cannot see any discussion of overshooting and envelope mixing in Fig. 4. Concerning Figs A35 and A36 full details should be given in the text or the captions on the modifications to the physics. Also, it is not clear whether the reference here is the corresponding model with classical PMS evolution without or with the assumed physics modification.

- p. 6, l. 275: presumably 'as' should be replaced by 'of'

- p. 9, caption to Fig. 3: This should make clear how the two 'predefined' stages are defined. The labelling of the panels is not sufficient.

- p. 12, 15, effects of rotation on the evolution and oscillations: It is clear that there is no fully physical description of rotation in 1D stellar modelling. However, at least the hydrostatic effect of the 20 % of critical rotation (assumed for the frequency calculation) could easily be taken into account in the 1D average approximation. Ref. 59 provides little information on the treatment of rotation in pulsation calculations and should be replaced. With the assumed rate, a perturbation treatment to second order would presumably be required.

- p. 12, l. 401: although I have little experience with MESA, I believe that the description of f_0 is incorrect. It is my understanding that $f_0 H_p$ is the distance below the edge of the unstable region where the convective diffusion coefficient $D_{\text{conv},0}$ is evaluated. This should be checked and, if needed, corrected.

- p. 12, l. 406: refs 50, 51, 52 have somewhat different compositions; which is actually used in the calculation?

- p. 12, l. 407: here the authors apparently introduce the trimer ' He_3 '. The ^3He isotope is presumably intended.

- p. 13, l. 428: There is no point in specifying the unit of the mass accretion here; the cgs value would in any case be used in the equation just above.

- p. 14, l. 446: this would be clearer as 'Here, M_{outer} defines the fractional mass of an outer region'

- Figs A1 - A34, A37: it would seem more logical to move the overview Figure A37 before Figs A1 - A34 providing detailed properties of the models.

- p. 42, caption to Fig. 38: Concerning the second panel, it should be stated that this is the mean frequency difference. The caption should refer to Figs A35 and A36 for details on the change in physics.

- p. 43, Fig. A39: It is not obvious from the figure or the text p. 15 how overshoot enters into this analysis. I note the interesting shift of the $n = 2$ mode in the later model, presumably reflecting some g-mode character and a change in the buoyancy frequency. This deserves some discussion in the text, particularly if the mode is in fact found to be excited. It is interesting that the p-mode (radius?) shift is more significant in the earlier than in the later model.

The meaning of the purple and green regions should be defined in the caption. Presumably the reference should be to Figure 4, rather than Figure 3?

Reviewer #2 (Remarks to the Author):

The manuscript entitled “A tale from the past: tracing the mass accretion history of stars with stellar pulsations” presents a very interesting approach to investigate the star formation in the pre-main sequence phase of a 2 solar mass star. Three different scenario’s of accretion are modelled and the effects on the stellar structure are investigated and presented in terms of differences that these cause in the observable frequencies. This is an important work, which I would recommend for publication after including the comments as listed below.

In this work a 2 Msol star (or better model) is used for the analysis and small frequency differences are obtained and compared with the frequency resolution of the state-of-the-art data. This is what could be measured if the mass, as well as the radius of a star is exactly known. In real life this is however not the case and we have uncertainties on the mass and radius. I think at least some mention of this should be made in the text. In the ideal case, it would be great to provide some numbers as to how accurate the mass and radius should be measured for this effect to be indeed visible.

In the same line of reasoning, the authors mention in the methods section that they tried different physics to see if there results are independent of the choice of physics. These results are presented in Figs A35 and A36, which are similar to Fig 4 (not Fig 3 as is referenced throughout the paper).

What I notice here is that there are differences in the frequency differences obtained (violin plots) of up to one magnitude. This seems in contradiction with the statement that the results are independent of the choice of physics and should be made more clear.

In general, I find the figures busy (which is not per se a problem) and in some cases not very well explained. For instance in Fig 4 there are two panels in the right bottom, which have the same label on the y-axis. If I understand the label correctly these panels show at each fractional radius the difference in fractional mass. What is the difference between the 2 panels? In the text that refers to this figure (lines 217-222) it is said that the radii of the disk-mediated accretion models are smaller than the classical model with a reference to the mass and Fig. 4, why not showing this in a more direct manner?

In Figure A38 in panel 2 the mean difference is written in the caption, while on the axis it is the absolute difference indicated. Furthermore there are 5 panels and only 4 are described. What is the difference between the fourth and the fifth panel?

Other question, why did the authors look at the $m=1, l=1$ modes and not at other modes? There should at least be a justification for that in the paper.

Throughout the paper, I found some places which seemingly contradict each other or need references: in the bold font part "It is further assumed that once the stars begin their fusion processes, they have forgotten their past" which seems to contradict with lines 52-54, where I also think that a reference should be added. Also the statement on 55 solar radii in line 57 could be strengthened with a reference.

In line 68 there is a comment on lower mass stars, which should be put in context.

Line 77-78: the sentence on the 'stellar birthline' could also do with a reference.

line 79: what is meant with 'inflated' initial conditions?

line 96: according to Aerts et al. 2010 (the book) delta Scuti stars can also be found in the H-shell burning phase, i.e. post main sequence. Why is that omitted here?

+++Additional comments:

In point #1 the referee asks the authors to not only show comparisons of the results at the same evolutionary phase, which is not an observable, and show it as a function of observables. This should

not be a lot of work and I would have expected at least a figure for the referee on this and depending on the finding an inclusion in the paper. Instead it is deemed beyond the scope of the paper, which I do not find a satisfactory answer.

For point #2 the authors do move most of the description of the main-sequence stars to the methods. However, they still leave it in the main part of the paper as well. I can understand that, though it comes across as a “half way house” and dilutes the message a bit.

The answer to #3 is satisfactory in the sense that this is the state of the art of current stellar models.

Regarding answer #4, the authors essentially say that what is discussed in the paper will indeed be very hard to observe due to lack of data for young stars and the sheer amount of oscillations observed in some of the main sequence delta scuti stars. So the answer to the comment of the referee is satisfactory, though I would have thought that it would be good to also include something in the paper stating that the predictions can only be tested with additional data, not currently available, and potentially a suggestion how to get these data.+++

Author: We are thankful to both referees for taking the time to review our manuscript and their fair and constructive criticism. The reviewers have addressed several points, with many of which we agree and for which we have applied major changes to the manuscript. This includes a proposed change to the manuscript title to better represent the science reported in the manuscript. We have replaced, removed and added several figures to address the reviewer's remarks. In addition, we have added additional explanations at several occasions and reworked the previous manuscript to make the content more reachable for the reader.

All changes are marked in brown colour in the revised manuscript. We further provide a point-by-point response to the referees' comments below.

Kind Regards,
Thomas Steindl (also on behalf of the co-authors)

Reviewer #1 (Remarks to the Author):

I thank the authors for taking, to a large extent, my comments on the previous version of the manuscript into account, in the resubmission. However, my main concern as to the suitability of the manuscript for a journal aimed for a broad readership remains: the manuscript in its title and abstract emphasizes the possibility of asteroseismic investigation of the detailed effects of the early evolution of stars, yet the claimed detectability relies on minute changes in the stellar radius at fixed evolutionary stages, characterized by the central abundances of carbon or hydrogen; I see no way that these points can be identified independently by other observations, allowing to detect the effect of the treatment of accretion. This concern is not addressed by the 'rebuttal' by the authors to the earlier referees' reports.

Author: Reading this statement from the referee, we propose a change to the manuscript title. While the referee thinks that “tracing the mass accretion history of stars with stellar pulsation” presents a basis for us to directly infer the mass accretion rate from the observed stellar pulsations, we do not propose this would be possible nor ever wanted to make any readers believe so. Hence to avoid any misunderstandings, we propose changing the title to “A tale from the past: the imprint of star formation on stellar pulsation”.

Regarding the suitability for a broad readership, we feel strongly opposed to the reviewer. We are regularly faced by the situation that researchers from various fields associate the pre-main sequence evolution with what we call the classical view, without questioning the validity of any assumptions made. In this regard, it is incredibly important to bring the complexity of the pre-main sequence evolution, with all its challenges, to a broad readership. This manuscript not only accomplishes this, but furthermore provides proof that the imprint of star formation is measurable long after accretion processes have stopped.

Concerning the results in the main text, as illustrated in Fig. 4, I note that only in the PMS model is the frequency shift significant, when the full set of models is considered. For the ZAMS model, the effect is essentially solely an increase in the theoretical uncertainty of the frequencies, with a slight bias.

Author: While this is true for the standard input physics presented in the main part, the effect is much larger when envelope mixing is increased to $5 \text{ cm}^2 \text{ s}^{-1}$ instead of $1 \text{ cm}^2 \text{ s}^{-1}$, which is still a relatively low value (compared to i.e. the SPB star KIC 8324482 for which asteroseismic modelling lead to an envelope mixing coefficient of

$\log D_{min} = 3.125_{-0.250}^{+0.125} \log(\text{cm}^2 \text{s}^{-1})$ <https://ui.adsabs.harvard.edu/abs/2020ApJ...899...38W/abstract>).

We have defined a standard input physics for which we present all our results. It would be simple for us to change the standard input physics to higher values of envelope mixing (since the values chosen by us are still comparably low) after seeing that the effect is larger in such a setup. While this would make the frequency differences larger, it would present a deliberate change of the hypothesis and should hence not be a step to undertake in particular regarding a strict scientific work ethic.

We have added information on the point raised by the referee to the manuscript to explain the frequency shift with respect to this concern. Additionally, we have added Figure 6 to the main body of the manuscript, presenting the frequency difference with the input physics including the higher envelope mixing.

As before, I do find the results interesting from the point of view of stellar evolution and hence, with suitable rewriting, for a specialized journal. Here it would also be interesting to relate the behaviour of the convective envelope, illustrated in Fig. 1 but perhaps analysed for a 1 Msun case, to the issues of differential composition differences, as discussed by Melendez et al. (2009; <https://ui.adsabs.harvard.edu/abs/2009ApJ...704L..66M/abstract>) and the preprint by Nordlund (<https://ui.adsabs.harvard.edu/abs/2009arXiv0908.3479N/abstract>). In support of such a revision, I have some additional comments:

Author: While the science case the referee describes is certainly interesting, we believe, with all due respect, that this is far out of the scope of this manuscript. As an inclusion would dilute the message the manuscript should deliver, we refrain from performing such an investigation as part of this submission and consider this an interesting topic for a further study.

- p. 3, l. 122 - 123: this is somewhat confusing: one would surely not describe a growth e-folding time as a growth rate. The growth rate is what is determined as the imaginary part of the frequency, from which the growth time (or, if you like, the growth e-folding time) can be calculated.

Author: In hindsight, we agree that the description was confusing and removed the remark to growth rate. The entire manuscript should now refer to growth e-folding times.

- p. 6, l. 240 - 241, Fig. 4: I cannot see any discussion of overshooting and envelope mixing in Fig. 4. Concerning Figs A35 and A36 full details should be given in the text or the captions on the modifications to the physics. Also, it is not clear whether the reference here is the corresponding model with classical PMS evolution without or with the assumed physics modification.

Author: This was a mistake of the previous version of the manuscript. Indeed, Figure 4 does not present results for the different mixing parameters. We understand that, at the previous state, this was confusing and are thankful for the reviewer for pointing this out. We have now added a description of the modification to the input physics for Figs. A35 and A36.

- p. 6, l. 275: presumably 'as' should be replaced by 'of'

Author: We believe the referee suggested to replace ‘evolution of accreting objects’ with ‘evolution as accreting objects’, which we gladly did. If indeed that was not the suggestion, we would like the referee to clarify his/her statement.

- p. 9, caption to Fig. 3: *This should make clear how the two 'predefined' stages are defined. The labelling of the panels is not sufficient.*

Author: We have added the definitions to the caption.

- p. 12, 15, *effects of rotation on the evolution and oscillations: It is clear that there is no fully physical description of rotation in 1D stellar modelling. However, at least the hydrostatic effect of the 20 % of critical rotation (assumed for the frequency calculation) could easily be taken into account in the 1D average approximation. Ref. 59 provides little information on the treatment of rotation in pulsation calculations and should be replaced. With the assumed rate, a perturbation treatment to second order would presumably be required.*

Author: Our setup follows the state of the art in asteroseismic calculations and should hence suffice for the purpose of this study. In our manuscript, we perform a study similar to the work of Michielsen et. al. (2019, <https://ui.adsabs.harvard.edu/abs/2019A%26A...628A..76M/abstract>) on SPB and Beta Cep stars. They, too apply the same approach concerning rotation. This approach is to compute the oscillation frequencies from non-rotating models. For the assumed rotation rate of 20% of critical rotation, this approach is valid, since the centrifugal acceleration and hence the flattening of the star can be ignored. The latter is possible until about ~50% of the critical velocity. Since the rotation frequencies are small compared to the pulsation frequencies, a first-order approach (Ledoux splitting) is adequate for the treatment of p-modes. Aerts et al. (2019, <https://ui.adsabs.harvard.edu/abs/2019A%26A...624A..75A/abstract>) showed that this is the case for SBP and Beta Cep stars which pulsate in lower frequencies compared to Delta Scuti stars which are discussed in this manuscript. The reference was indeed to the wrong article and we are thankful to the reviewer for pointing this out. We have added a discussion towards the end of “Methods: Stellar pulsation frequencies” and replaced the incorrect reference.

- p. 12, l. 401: *although I have little experience with MESA, I believe that the description of f_0 is incorrect. It is my understanding that $f_0 H_p$ is the distance below the edge of the unstable region where the convective diffusion coefficient $D_{conv,0}$ is evaluated. This should be checked and, if needed, corrected.*

Author: We cite the reference for MESA (<https://docs.mesastar.org/en/release-r21.12.1/reference/controls.html>) Although the reference is for a newer version, the definition has not changed, but the explanation is superior to the older version.

“The switch from convective mixing to overshooting happens at a distance $f_0 H_p$ into the convection zone from the estimated location where $\text{grad}_{ad} = \text{grad}_{rad}$, where H_p is the pressure scale height at that location. A value ≤ 0 for f_0 is a mistake – you are required to set f_0 as well as f . take a look at the following from an email concerning this: Overshooting works by taking the diffusion mixing coefficient at the edge of the convection zone and extending it beyond the zone. But – and here’s the issue – at the exact edge of the zone the mixing coefficient goes to 0. So we don’t want that. Instead we want the value of the mixing

coeff NEAR the edge, but not AT the edge. The “f0” parameter determines the exact meaning of “near” for this. It tells the code how far back into the zone to go in terms of scale height. The overshooting actually begins at the location determined by f0 back into the convection zone rather than at the edge where the diffusion coeff is ill-defined. So, for example, if you want overshooting of 0.2 scale heights beyond the normal edge, you might want to back up 0.05 scale heights to get the diffusion coeff from near the edge and then go out by 0.25 scale heights from there to reach 0.2 Hp beyond the old boundary. In the inlist this would mean setting the “f0” to 0.05 and the “f” to 0.25.”

We have additionally rephrased the sentence slightly for more clarity.

- p. 12, l. 406: refs 50, 51, 52 have somewhat different compositions; which is actually used in the calculation?

Author: We are using the predefined metal abundances corresponding to (initial_zfracs = 8) and the original description was according to the standard in the literature. We follow the wish of the referee for a more detailed description and have now clearly defined which values are taken for that specific predefined MESA choice.

- p. 12, l. 407: here the authors apparently introduce the trimer 'He_3'. The ^3He isotope is presumably intended.

Author: We followed the referee’s suggestion and applied these changes.

- p. 13, l. 428: There is no point in specifying the unit of the mass accretion here; the cgs value would in any case be used in the equation just above.

Author: We followed the referee’s suggestion and applied these changes.

- p. 14, l. 446: this would be clearer as 'Here, M_{outer} defines the fractional mass of an outer region'

Author: We followed the referee’s suggestion and applied these changes.

- Figs A1 - A34, A37: it would seem more logical to move the overview Figure A37 before Figs A1 - A34 providing detailed properties of the models.

Author: The reason for this was solely the order of which the figures were referenced in the text. We have referenced the overview Figure (previously A37, now A1) in the main text and can hence change the order to follow the referee’s suggestion.

- p. 42, caption to Fig. 38: Concerning the second panel, it should be stated that this is the mean $_frequency_difference$. The caption should refer to Figs A35 and A36 for details on the change in physics.

Author: We have now decided to drop the discussion of the main sequence results altogether (also in response to the second referee’s remarks) and this Figure is hence no longer part of the submitted manuscript.

- p. 43, Fig. A39: It is not obvious from the figure or the text p. 15 how overshoot enters into this analysis. I note the interesting shift of the $n = 2$ mode in the later model, presumably reflecting some g-mode character and a change in the buoyancy frequency. This deserves some discussion in the text, particularly if the mode is in fact found to be excited. It is interesting that the p-mode (radius?) shift is more significant in the earlier than in the later model.

The meaning of the purple and green regions should be defined in the caption. Presumably the reference should be to Figure 4, rather than Figure 3?

Author: We chose to present the model with enhanced overshooting coefficients as the frequency differences are larger compared to the standard input physics. Our results indeed suggest that this mode reflects g-mode characteristics and agree with the referee that there should have been a more detailed discussion. However, we have now decided to drop the discussion of the main sequence results altogether (also in response to the second referee's remarks) and hence will not present any discussion of these results in the reworked manuscript.

Reviewer #2 (Remarks to the Author):

The manuscript entitled "A tale from the past: tracing the mass accretion history of stars with stellar pulsations" presents a very interesting approach to investigate the star formation in the pre-main sequence phase of a 2 solar mass star. Three different scenario's of accretion are modelled and the effects on the stellar structure are investigated and presented in terms of differences that these cause in the observable frequencies. This is an important work, which I would recommend for publication after including the comments as listed below.

In this work a 2 Msol star (or better model) is used for the analysis and small frequency differences are obtained and compared with the frequency resolution of the state-of-the-art data. This is what could be measured if the mass, as well as the radius of a star is exactly known. In real life this is however not the case and we have uncertainties on the mass and radius. I think at least some mention of this should be made in the text. In the ideal case, it would be great to provide some numbers as to how accurate the mass and radius should be measured for this effect to be indeed visible.

Author: For asteroseismic modelling to be applicable, the stellar mass and radius does not have to be known. As demonstrated by multiple authors including Pedersen et al. (2021, Nature Astronomy, <https://ui.adsabs.harvard.edu/abs/2021NatAs...5..715P/abstract>) it is sufficient to know the parameters T_{eff} , $\log g$, $[M/H]$, and $\log(L/L_{\odot})$. A set of theoretical models that fulfills this condition can hence be constructed. Asteroseismic modelling then relies on a statistical approach, ideally based upon a multivariate regression model (see Aerts et al. 2018, <https://ui.adsabs.harvard.edu/abs/2018ApJS..237...15A/abstract>) and according to the work references the use of a Mahalanobis distance. While this asteroseismic modelling technique was discussed for the application to g-mode period spacing, a similar technique is applicable for pressure mode pulsations as well. Today's state of the art investigation of mixing profiles in g-mode pulsating stars is based on a work that has also been the motivation for this manuscript (Michielsen et al. 2019,

<https://ui.adsabs.harvard.edu/abs/2019A%26A...628A..76M/abstract>) which presents the influence of the mixing profiles on the g-mode frequencies.

To make this more reachable for the reader, we have edited the discussion towards the end of the main manuscript and added ‘forward asteroseismic modelling of pre-main sequence δ -Scuti star’ stars to methods in which we discuss the above.

In the same line of reasoning, the authors mention in the methods section that they tried different physics to see if there results are independent of the choice of physics. These results are presented in Figs A35 and A36, which are similar to Fig 4 (not Fig 3 as is referenced throughout the paper). What I notice here is that there are differences in the frequency differences obtained (violin plots) of up to one magnitude. This seems in contradiction with the statement that the results are independent of the choice of physics and should be made more clear.

Author: The statement that our results are independent of the choice of input physics was regarding the difference between the classical and disk-mediated model to be larger than the expected uncertainties, which is the case. In hindsight however, we agree with the referee’s remark that this needs to be phrased clearer. To reach that goal, we have added more discussion at the end of “Methods: Stellar Evolution Models” and refer to that in the main manuscript.

In general, I find the figures busy (which is not per se a problem) and in some cases not very well explained. For instance in Fig 4 there are two panels in the right bottom, which have the same label on the y-axis. If I understand the label correctly these panels show at each fractional radius the difference in fractional mass. What is the difference between the 2 panels? In the text that refers to this figure (lines 217-222) it is said that the radii of the disk-mediated accretion models are smaller than the classical model with a reference to the mass and Fig. 4, why not showing this in a more direct manner?

Author: We agree with the reviewer that especially Figure 4 is very busy, partly because of the many different panels. To address this issue, we have split the former Figure 4 into two Figures (now Figures 4 and 5). The new Figure 4 only shows the Hertzsprung Russel diagram and the frequency differences as violin plots, and the new Figure 5 shows the differences in stellar structure for the 34 disk-mediated accretion models. We have additionally added a Figure 6, which presents the frequency difference for the input physics with enhanced envelope mixing to focus on the important change in pulsation frequency when moving from classical to disk-mediated models.

In Figure A38 in panel 2 the mean difference is written in the caption, while on the axis it is the absolute difference indicated. Furthermore there are 5 panels and only 4 are described. What is the difference between the fourth and the fifth panel?

Author: The referee is correct, that the labels for the first two axes should indicate the mean, but did not do so. However, as we have removed the discussion of main sequence stars from the manuscript entirely, Figure A38 is no longer part of the submission.

Other question, why did the authors look at the $m=1, l=1$ modes and not at other modes? There should at least be a justification for that in the paper.

Author: We agree with the reviewer that there is no justification in the manuscript. We have added an explanation that dipole and radial modes are the most commonly observed pulsation frequencies in delta Scuti stars. We have done additional calculations for radial pulsation modes but the results for the frequency shifts are virtually identical. We attach here the corresponding plots of Figure 2 and Figure A36, but for radial modes. As is evident, the values are almost identical, hence we refrain from adding additional plots to the manuscript but discuss this result.

Figure A36, but with radial modes.

Figure 2 but with radial modes.

Throughout the paper, I found some places which seemingly contradict each other or need references: in the bold font part “It is further assumed that once the stars begin their fusion processes, they have forgotten their past” which seems to contradict with lines 52-54, where I also think that a reference should be added. Also the statement on 55 solar radii in line 57 could be strengthened with a reference.

Author: We are more than sorry but cannot seem to find a contradiction between “It is further assumed that once the stars begin their fusion processes, they have forgotten their past” and lines 52-54 which state “However, our general theory of early stellar evolution lacks essential physical ingredients that are either not well understood or not accurately included in our theoretical models.”.

In hope that this addresses the reviewers remark, we changed the bold font text to “In this inaccurate picture, the common belief is that once the stars begin their fusion processes, they have forgotten their past” in an effort to make it clear that this is often the belief of non-specialists about the early part of stellar evolution.

If this answer is not satisfactory to the reviewer, we would kindly ask to clarify his/her remark.

The statement on 55 solar radii relates to the pre-main sequence model created with the standard routine in MESA, an information that was now added to the manuscript as footnote because we believe it would otherwise impair the readability too much.

In line 68 there is a comment on lower mass stars, which should be put in context.

Author: The comment regarding the pp-chain for lower mass stars was a request by a reviewer. We have now added additional information to the comment in the hope that this puts this comment better in context.

Line 77-78: the sentence on the ‘stellar birthline’ could also do with a reference.

Author: We have added a reference to Baraffe et al. 2009 (<https://ui.adsabs.harvard.edu/abs/2009ApJ...702L..27B/abstract>) who state in their abstract “Our results also show that the concept of a stellar birthline for low-mass objects has no valid support.”

line 79: what is meant with ‘inflated’ initial conditions?

Author: This was in reference to the huge initial radius, the classical pre-main sequence model obtained from the standard routines. We agree that “inflated” is not the best word to describe this and changed the wording to “simplified”.

line 96: according to Aerts et al. 2010 (the book) delta Scuti stars can also be found in the H-shell burning phase, i.e. post main sequence. Why is that omitted here?

Author: We have added a comment about post-main sequence delta Scuti stars and the reference to Aerts et al. 2010.

+++Additional comments:

In point #1 the referee asks the authors to not only show comparisons of the results at the same evolutionary phase, which is not an observable, and show it as a function of observables. This should not be a lot of work and I would have expected at least a figure for the referee on this and depending on the finding an inclusion in the paper. Instead it is deemed beyond the scope of the paper, which I do not find a satisfactory answer.

Author: In our revised manuscript, we have followed the initial comment by the reviewer and compare the different models at the same radius. We have defined a radius of 2.8 solar radii at which we have saved the models at an evolutionary stage after the stars have entered the expected instability region for pre-main sequence delta Scuti stars and show the results in Figure A38.

For point #2 the authors do move most of the description of the main-sequence stars to the methods. However, they still leave it in the main part of the paper as well. I can understand that, though it comes across as a “half way house” and dilutes the message a bit.

Author: In light of the referee’s remark we have now completely removed our results concerning the main sequence from the manuscript. The entire manuscript is now focused on the pre-main sequence and the very early main-sequence (which is the pre-defined ZAMS point) in a hope to strengthen the message of the main work.

The answer to #3 is satisfactory in the sense that this is the state of the art of current stellar models.

Regarding answer #4, the authors essentially say that what is discussed in the paper will indeed be very hard to observe due to lack of data for young stars and the sheer amount of oscillations observed in some of the main sequence delta scuti stars. So the answer to the comment of the referee is satisfactory, though I would have thought that it would be good to also include something in the paper stating that the predictions can only be tested with additional data, not currently available, and potentially a suggestion how to get these data.+++

Author: We agree with the reviewer that the explanation in the manuscript regarding this point are insufficient and have added additional explanations towards the end of the main manuscript.

REVIEWER COMMENTS

Reviewer #1 (Remarks to the Author):

I thank the authors for reacting positively and in some detail to my earlier comments, and essentially taking full account of them. I am certainly happy to have been able to help correcting some problems with the earlier version. Concerning the specific point p. 6, l. 275 (now l. 291) I can confirm that what I thought was an obvious mistake has been correctly dealt with. I also agree that the revised title makes clearer what the manuscript aims to accomplish, thus making it more appropriate for this journal.

Let me, however, already now note that the caption to Fig. 2 has not been consistently updated to reflect the introduction of the labelling of the panels. $l = 1$ should be set as an inline equation in the caption to Fig. 2. This should be carefully corrected, making also clear which of the panels refer to the classical, respectively mass-accretion, models.

Beyond this I have just a few minor comments:

p. 4, l. 189: Change 'A1' to 'A2'

p. 6, l. 245: 'this result makes' or 'these results make'

p. 6, l. 248: 'lead' -> 'leading'

p. 18, l. 537: 'different to' -> 'different from'

p. 19, l. 607: 'various days' looks very odd.

I clearly cannot guarantee to have caught all such smaller issues, and hence I recommend a careful read-through by all the authors before the final submission.

Reviewer #2 (Remarks to the Author):

The authors have included my comments generally speaking to a satisfying level. A few points I still want to make:

I agree with the authors that we don't have to know M and R , and other parameters are used. However, my comment was meant in a more general way: how accurate do we need to know other stellar parameters to be able to use these differences in frequency to constrain the accretion history.

Regarding my recommendation for publication: I do think this is scientifically solid work and could henceforth be published. If it is exciting enough given my comment above for Nature Communications, that I can't judge.

Author: We are thankful to both referees for again taking the time to review our manuscript. Your remarks throughout the review process have improved the manuscript a great deal and we are hopeful that the newly revised version can clear any remaining concerns.

Kind Regards,
Thomas Steindl (also on behalf of the co-authors)

Reviewer #1 (Remarks to the Author):

I thank the authors for reacting positively and in some detail to my earlier comments, and essentially taking full account of them. I am certainly happy to have been able to help correcting some problems with the earlier version. Concerning the specific point p. 6, l. 275 (now l. 291) I can confirm that what I thought was an obvious mistake has been correctly dealt with. I also agree that the revised title makes clearer what the manuscript aims to accomplish, thus making it more appropriate for this journal.

Let me, however, already now note that the caption to Fig. 2 has not been consistently updated to reflect the introduction of the labelling of the panels. $l = 1$ should be set as an inline equation in the caption to Fig. 2. This should be carefully corrected, making also clear which of the panels refer to the classical, respectively mass-accretion, models.

Author: We are thankful to the reviewer for pointing towards this inconsistency. In our revised manuscript, we have reworked the figure caption to use the labelled panels only. In addition, we have added a description of the evolutionary phase the panels refer to and hope that the description of Figure 2 is now clear to any reader.

Beyond this I have just a few minor comments:

p. 4, l. 189: Change 'A1' to 'A2'

p. 6, l. 245: 'this result makes' or 'these results make'

p. 6, l. 248: 'lead' -> 'leading'

p. 18, l. 537: 'different to' -> 'different from'

p. 19, l. 607: 'various days' looks very odd.

I clearly cannot guarantee to have caught all such smaller issues, and hence I recommend a careful read-through by all the authors before the final submission.

Author: We are thankful to the reviewer for pointing out these mistakes which we have gladly corrected in our manuscript. The sentence at p. 19, l. 607 has been rephrased including some more details. Before resubmission, all authors have had a careful read-through the manuscript and we are hopeful to have caught all such smaller issues.

Reviewer #2 (Remarks to the Author):

The authors have included my comments generally speaking to a satisfying level. A few points I still want to make:

I agree with the authors that we don't have to know M and R , and other parameters are used. However, my comment was meant in a more general way: how accurate do we need to know other stellar parameters to be able to use these differences in frequency to constrain the accretion history.

Author: Following the methodology of Aerts et al. (2018, <https://ui.adsabs.harvard.edu/abs/2018ApJS..237...15A/abstract>) the spectroscopic parameters are not included in the modelling approach but are treated as a posteriori evaluation regarding how good the model is. The same is true for i.e. the excitation of modes. For pulsation frequencies extracted from a light curve sufficiently long enough for our application, the observed frequencies $\epsilon_i/f_i \ll 0.1\%$ are far more precisely known than any of the stellar parameters, i.e. $\log g$ or $\log T_{\text{eff}}$.

In the practical case, $\log g$ and $\log T_{\text{eff}}$ are mostly used to determine the range of initial parameters to produce the initial model grid or used a posteriori to eliminate stellar models lying outside the observed regime, i.e. the “spectroscopic error box” (see e.g. Buyschaert et al. 2018 <https://ui.adsabs.harvard.edu/abs/2018A%26A...616A.148B>, Michielsen et al. 2021 <https://ui.adsabs.harvard.edu/abs/2021A%26A...650A.175M>, Pedersen et al. 2021 <https://ui.adsabs.harvard.edu/abs/2021NatAs...5..715P>).

Given that the calculation time for disk-mediated pre-main sequence models is comparably long, the calculation of a model grid will be very time consuming. In this regard, the accuracy of the known stellar parameters can be important whether such a modelling approach is feasible given specific time constraints. For pre-main sequence δ -Scuti stars, the observation of an echelle diagram usually provides narrow constraints on $\log g$, but far less rigid constraints on the surface temperature. For a sufficiently reduced model grid, an accurate knowledge of the surface temperature and metallicity would be very beneficial.

However, at the current stage we do not assume that a higher than usual accuracy on any stellar parameter (for example ± 150 K in T_{eff} and ± 0.1 in $\log g$ in the mass regime discussed here) would be needed to constrain the models as they do not influence the Mahalanobis distance used as merit function.

We have added a respective discussion to section “Methods: Forward asteroseismic modelling of pre-main sequence δ -Scuti stars”.

Regarding my recommendation for publication: I do think this is scientifically solid work and could henceforth be published. If it is exciting enough given my comment above for Nature Communications, that I can't judge.

REVIEWERS' COMMENTS

Reviewer #1 (Remarks to the Author):

I am happy that the authors found useful these final comments, and they have been appropriately implemented.

I have no further comments.

Reviewer #2 (Remarks to the Author):

I would like to thank the authors for carefully looking at the comments I made. My only response / surprise is that the implication of the statements made that “models can be constraint with only the frequencies”, there is apparently no other physics that can mimic these changes in the frequencies between models with different PMS scenarios, and hence the models are not degenerate. I am not an expert on δ -Scuti stars in particular, and this was not within my expectations.

The authors have included my comments generally speaking to a satisfying level. A few points I still want to make:

I agree with the authors that we don't have to know M and R , and other parameters are used. However, my comment was meant in a more general way: how accurate do we need to know other stellar parameters to be able to use these differences in frequency to constrain the accretion history.

Author: Following the methodology of Aerts et al. (2018, <https://ui.adsabs.harvard.edu/abs/2018ApJS..237...15A/abstract>) the spectroscopic parameters are not included in the modelling approach but are treated as a posteriori evaluation regarding how good the model is. The same is true for i.e. the excitation of modes. For pulsation frequencies extracted from a light curve sufficiently long enough for our application, the observed frequencies $\epsilon_i/f_i \ll 0.1\%$ are far more precisely known than any of the stellar parameters, i.e. $\log g$ or $\log T_{\text{eff}}$.

In the practical case, $\log g$ and $\log T_{\text{eff}}$ are mostly used to determine the range of initial parameters to produce the initial model grid or used a posteriori to eliminate stellar models lying outside the observed regime, i.e. the “spectroscopic error box” (see e.g. Buysschaert et al. 2018 <https://ui.adsabs.harvard.edu/abs/2018A%26A...616A.148B>, Michielsen et al. 2021 <https://ui.adsabs.harvard.edu/abs/2021A%26A...650A.175M>, Pedersen et al. 2021 <https://ui.adsabs.harvard.edu/abs/2021NatAs...5..715P>).

Given that the calculation time for disk-mediated pre-main sequence models is comparably long, the calculation of a model grid will be very time consuming. In this regard, the accuracy of the known stellar parameters can be important whether such a modelling approach is feasible given specific time constraints. For pre-main sequence δ -Scuti stars, the observation of an echelle diagram usually provides narrow constraints on $\log g$, but far less rigid constraints on the surface temperature. For a sufficiently reduced model grid, an accurate knowledge of the surface temperature and metallicity would be very beneficial.

However, at the current stage we do not assume that a higher than usual accuracy on any stellar parameter (for example ± 150 K in T_{eff} and ± 0.1 in $\log g$ in the mass regime discussed here) would be needed to constrain the models as they do not influence the Mahalanobis distance used as merit function.

We have added a respective discussion to section “Methods: Forward asteroseismic modelling of pre-main sequence δ -Scuti stars”.

Regarding my recommendation for publication: I do think this is scientifically solid work and could henceforth be published. If it is exciting enough given my comment above for Nature Communications, that I can't judge.

Author: We are thankful to the referee for again taking the time to review our manuscript. Your remarks throughout the review process have improved the manuscript a great deal and we are happy that we could clear the last remaining concerns.

Kind Regards,
Thomas Steindl (also on behalf of the co-authors)

Reviewer #2 (Remarks to the Author):

I would like to thank the authors for carefully looking at the comments I made. My only response / surprise is that the implication of the statements made that “models can be constraint with only the frequencies”, there is apparently no other physics that can mimic these changes in the frequencies between models with different PMS scenarios, and hence the models are not degenerate. I am not an expert on δ -Scuti stars in particular, and this was not within my expectations.

Author: We believe the statement “models can be constraint with only the frequencies” is too strong. The classical parameters deliver important constraints. What we have described in our past correspondence is the methodology developed by Aerts et al. (2018) for which only the pulsation frequencies are included in the calculation of the merit function, the Mahalanobis distance. This is similar in other modelling approaches applied in the literature that make use of a simple χ^2 function and apply a simple cut off with the spectroscopic error box as has been performed in i.e. Murphy et al. 2021 (<https://ui.adsabs.harvard.edu/abs/2021MNRAS.502.1633M/abstract>). In comparison to the latter, the approach of forward asteroseismic modelling described by Aerts et al. (2018) is to be preferred and we hope to apply it in the future. Without accurate knowledge of the classical constraints, however, it is highly unlikely to arrive at the position for which forwards asteroseismic modelling can be performed in the first place. We have added a paragraph at the end of the main text and rephrased part of the ‘Forward asteroseismic modelling’ section towards the importance of classical constraints when describing the methods.